# Genetics of Oocyte Maturation Defects and Early Embryo Development Arrest

**DOI:** 10.3390/genes13111920

**Published:** 2022-10-22

**Authors:** Olga Aleksandrovna Solovova, Vyacheslav Borisovich Chernykh

**Affiliations:** 1Moscow Regional Research and Clinical Institute (MONIKI), Schepkina st. 61/2, 129110 Moscow, Russia; 2Research Centre for Medical Genetics, Moskvorechie St. 1, 115522 Moscow, Russia

**Keywords:** embryo, female infertility, fertilization, genes, human reproduction, oocyte, oogenesis

## Abstract

Various pathogenic factors can lead to oogenesis failure and seriously affect both female reproductive health and fertility. Genetic factors play an important role in folliculogenesis and oocyte maturation but still need to be clarified. Oocyte maturation is a well-organized complex process, regulated by a large number of genes. Pathogenic variants in these genes as well as aneuploidy, defects in mitochondrial genome, and other genetic and epigenetic factors can result in unexplained infertility, early pregnancy loss, and recurrent failures of IVF/ICSI programs due to poor ovarian response to stimulation, oocyte maturation arrest, poor gamete quality, fertilization failure, or early embryonic developmental arrest. In this paper, we review the main genes, as well as provide a description of the defects in the mitochondrial genome, associated with female infertility.

## 1. Introduction

Infertility is a global problem that affects 48 million couples [1]. The female factor accounts for approximately 37% of infertility cases [2]. The percentage of couples with reproductive problems is increasing in number, and more and more patients are turning to reproduction centers for assisted reproduction technologies (ART). With the use of ART, especially controlled ovarian stimulation, in vitro fertilization (IVF), and intracytoplasmic sperm injection (ICSI) techniques, many infertile women have a chance to become a pregnant using their own oocytes. However, a large number of women cannot become pregnant because of recurrent failures of IVF and/or ICSI programs due to poor response to ovarian stimulation, oocyte maturation arrest (OMA), poor quality of gametes, fertilization failure, or early embryo developmental arrest [3]. 

Many cases of female infertility result from problems connected to producing eggs. This can be caused by various factors (or their combination), including defects in gonadal differentiation and ovary development, folliculogenesis, oocyte maturation and gamete recognition, or early embryonic development. Genetic, endocrine, surgical, and anatomical abnormalities of the reproductive system can affect human fertility. The genetic factors of reproduction disorders include various chromosomal abnormalities (aneuploidy, balanced and unbalanced chromosome rearrangements, mosaicism), copy number variations (CNVs) and pathogenic variants in multiple genes that are responsible for female infertility [4].

Besides chromosomal abnormalities, CNVs and gene variants, there are other genetic factors that affect folliculogenesis and oogenesis, oocyte quality, and embryo competence, including the mitochondrial genome (mtDNA copy number and mutations), various non-coding RNAs, and epigenetic factors [5]. However, the genetic basis for female infertility associated with meiotic defects, oocytes/embryo aneuploidy, abnormalities in oocyte development, fertilization, and early embryogenesis is not well understood.

## 2. Oogenesis, Meiosis, and Oocyte Maturation

In humans, the early gonad development in females and males is identical within first 4 weeks of gestation. Before sex differentiation and at the early stages of gonadogenesis, mitotic proliferated primordial germ cells, PGCs (gonocytes), migrate in the gonadal ridges [6]. The bipotential, sex-indifferent gonadal ridge forms as a thickening of the coelomic or surface epithelium on ventral side of a mesonephros. Sex differentiation in male or female pathways begins at fifth week of the gestation, at that ovarian development occurs in the absence of activation of male-type sex differentiation and depends on expression of several transcription factors (*DAX-1*, *FIGLA*, *FOXL2*, *GATA4*, *LHX8*, *LHX9*, *NOBOX*, *NR5A1/SF1*, *SOHLH1*, *SOHLH2*, *SOX9*, *WNT4*, and *WT1*) [7].

During the gonadotropin-independent stage of oogenesis, ovary development begins and post-migratory gonocytes interact with the differentiating somatic cells of ovarian tissue and differentiate into oogonia, which enter the process of meiosis, forming primordial follicles (Figure 1) [8]. In mammalians, multiple genes are involved in spermatogenesis and oogenesis, meiosis, and gametes maturation. Studies on mice show that *Sohlh1* and *Sohlh2* genes can affect both male and female germ cell differentiation, while *Figla, Lhx8*, and *Nobox* are female germ cell-specific genes and do not affect spermatogenesis [9]. These studies have uncovered a number of genes that can lead to oocyte maturation defects and female infertility, for example, *Cdc25b* (OMIM: 116949), *Marf1* (OMIM: 614593), and *Pde3a* (OMIM: 123805). In humans, heterozygous and biallelic pathogenic variants in *FIGLA, FOXL2, NOBOX, NR5A1/SF1, SOHLH1, SOHLH2, WNT4*, and *WT1* genes have been detected in female patients with a wide spectrum of abnormalities of reproductive system, including non-syndromic (“isolated”) and syndromic diseases and reproductive disorders associated with female infertility, for example, premature ovarian failure/insufficiency (POF/POI), disorders of sex development (DSD), and, in particular, complete and incomplete gonadal dysgenesis [7,10].

Gametogenesis is the most important component of both male and female fertility. During meiosis, both immature germ cells undergo two divisions, meiosis I and II, and as a result, haploid (1n) gametes are formed from diploid (2n) cells (Figure 1). In prophase I of meiosis, genetic recombination occurs between homologous (paternal and maternal) chromosomes. After the diplotene of prophase I, oocytes are arrested (germinal vesicle, GV oocytes). Before puberty, the oocytes are maintained at a cellular quiescent stage of prophase I, termed “dictyotene arrest” (GV stage arrest). Dictyotene-arrested oocytes resume meiosis I upon stimulation by a luteinizing hormone (LH) and meiosis I, which is completed by extruding the first polar body (PB). During puberty, a gonadotropin-dependent stage of folliculogenesis occurs, and the release of luteinizing hormone (LH), secreted by the adenohypophysis, resumes meiosis. Chromatin condensation, the breakdown of the nuclear envelope, spindle formation, and chromosome alignment occurs in oocytes, and finally, meiosis I is completed upon the extrusion of the first polar body (PB1). After that oocytes enter meiosis II and arrest at metaphase II (MII) until fertilization. Meiosis II is fully completed only after the fertilization of the oocyte by a sperm (Figure 1) [11,12].

Various defects in meiosis commonly result in abnormal chromosome segregation and are some of the causes of oocyte maturation arrest. Meiotic defects in oogenesis can lead to female infertility because of oocyte maturation arrest and increased aneuploidy and the poor quality and morphology of female gametes and embryos, which often results in low rates of successful IVF/ICSI procedures and early pregnancy loss.

## 3. Maternal Effect Genes (MEGs)

Recently, a new Mendelian disease associated with oocyte maturation arrest (especially OMIM: 616780 and OMIM: 617743), human embryonic lethality at an earlier stage of development, and some genes responsible for the corresponding phenotypes were discovered. Several genes have been found to be involved in the etiology of oocyte maturation arrest (*TUBB8*, OMIM: 616768; *PATL2*, OMIM: 614661) and fertilization failure (*TLE6*, OMIM: 612399; *WEE2*, OMIM: 614084) [13,14,15,16] (Table 1).

Maternal-effect genes (MEGs) are transcribed in female germ cell and their mRNA influences the development of oocytes and embryos [17]. One of the most frequent genes leading to impaired oocyte maturation and embryonic development is the *TUBB8* gene.

***TUBB8*** (tubulin, beta 8 class VIII) is located on 10p15.3. It consists of four exons and encodes for a 444-amino acid protein. *TUBB8* codes primate-specific β-tubulin isotype, mainly expressed at different stages of human oocyte and the early embryo divisions, and is absent in mature spermatozoa [16].

Tubulin (α/β) is the structural unit of microtubules. Microtubules play a crucial role in spindle formation during meiosis [18]. Therefore, alterations in the TUBB8 protein structure associated with oocyte maturation defects and early embryo developmental arrest. More than 130 variants have been reported in *TUBB8* gene [16,19,20,21,22,23,24]. Pathogenic variants in *TUBB8* gene alter the microtubule behavior and oocyte meiotic spindle assembly. In female carriers, clinical phenotypes include oocyte maturation arrest, fertilization failure, early embryo developmental arrest, and the failure of implantation. It has been reported that variants in the *TUBB8* gene could be the cause of zygotic arrest in some embryos by disrupting the process of cell division [16]. Heterozygous pathogenic variants in *TUBB8* in most cases result in metaphase I (MI) arrest due to a dominant negative effect, although arrest at early embryonic development has also been described [23]. Homozygous or compound-heterozygous variants associated with milder phenotypes, including fertilization failure (MII arrest), early embryonic developmental arrest, and zygotic cleavage failure [25,26]. Conducted functional studies confirm the severity of dominant variants on oocyte development [16,27]. These data indicate that different variants can lead to different structural abnormalities and can differently disrupt the interaction of TUBB8 with kinesin or its binding with other proteins that are the part of the microtubule structure [19]. Functional studies showed that pathogenic variants in *TUBB8* gene affect chaperone-dependent folding and the assembly of the α/β-tubulin heterodimer, disturb the microtubule structure and further lead to spindle-assembly defects and oocyte maturation arrest [16].

Pathogenic variants in the *TUBB8* and *PATL2* gene cause oocyte metaphase I (MI) stage arrest (OMIM: 616780). *TUBB8* is the most common gene associated with oocyte maturation defects with its pathogenic variants detected in approximately 30% of patients with OMA [16]. However, in the most cases, the genetic cause of OMA remains unknown.

The ***CDC20*** (*cell division cycle 20*) gene, located in 1p34.2, consists of 11 exons and encodes for a 499-amino acid protein [28]. CDC20 is expressed in many tissues, including germ cells. CDC20 protein is a co-activator of anaphase-promoting complex/cyclosome (APC/C), required for mitotic and meiotic escape. CDC20 molecule can adheres to APC/C, forming a complex APC/CCDC20, which promotes the destruction of securin and cyclin B, the inactivation of the cyclin-dependent kinase-1 and -2 (CDK1 and CDK2), and the separation of sister chromatids and, thus, provides the normal transition from metaphase to anaphase [29,30]. During the cell cycle, the highest level of CDC20 expression is detected at the G2/M stage [31].

Biallelic pathogenic variants in the *CDC20* gene are responsible for prolonged human oocyte maturation arrest, fertilization failure, and early embryonic development arrest [32,33]. The homozygous or compound-heterozygous variants of *CDC20* are also associated with male infertility resulted from non-obstructive azoospermia [34]. Several studies show that different types of *CDC20* variants lead to the development of different phenotypes, including defects in oocyte maturation, fertilization failure, and embryonic development arrest [32,33]. It has been reported that the injection of CDC20-coding RNA into the oocytes of patients could recuse the phenotype of MI arrest [32,33].

The ***TRIP13*** (*thyroid hormone receptor interactor 13*) gene is located on 5p15.33. It consists of 13 exons and encodes for a 432-amino acid protein, AAA-ATPase, which is a key component of the spindle assembly checkpoint (SAC) [35]. *TRIP13* gene plays a crucial role in mitosis and meiosis, and it is expressed in many tissues, including germ cells [35]. During mitosis, TRIP13 protein supports spindle checkpoint silencing, and in meiosis, TRIP13 is involved in recombination pathways [36]. Different types of *TRIP13* recessive pathogenic variants cause distinct diseases, including Wilms tumors and female infertility through the different effects on mitosis and meiosis [37,38]. Homozygous nonsense variants or pathogenic variants affecting the splicing process associated with a total loss of TRIP13 protein function and an abnormal mitosis, whereas missense pathogenic variants are only affected oocyte meiosis with a mild decrease protein production [39]. Homozygous and compound-heterozygous pathogenic *TRIP13* missense variants are associated with female infertility, mostly characterized by oocyte meiotic arrest and abnormal zygote cleavage [38]. Injecting TRIP13-coding RNA into the oocytes of affected females resulted in the restoration of function, which has implications for future therapeutic treatment [38].

The ***PATL2*** (*protein associated with topoisomerase II*) gene is located on 15q21.1. It consists of 15 exons and encodes for a highly conserved oocyte-specific mRNP repressor of translation, which regulates the expression of many genes involved in oocyte maturation and early embryonic development [40]. The *PATL2* gene is specifically expressed in immature oocytes, and as the oocyte matures, the expression of *PATL2* greatly decreases [13].

Homozygous or compound-heterozygous *PATL2* variants lead to infertility due to oocyte germinal vesicle (GV) arrest or MI arrest, fertilization failure, and early embryo developmental arrest [13,40,41]. The effect of *PATL2* is limited to females; males with a homozygous of a severe truncating variant are fertile [42].

Researchers have observed the disturbance of oocyte maturation, the morphological defects of polar body, and abnormal spindle assembly after the microinjection of corresponding mutated mRNA [41].

The ***BTG4*** (*B-cell translocation gene 4*) gene is located on 11q23.1. It consists of five exons and encodes for a 223-amino acid protein. BTG4 is predominantly expressed in oocytes, ovaries, and early embryos [43]. BTG4 protein interacts with the CCR4-NOT complex, which promotes the degradation of maternal mRNAs and accelerates the maternal-to-zygotic transition (MZT) by promoting mRNA deadenylation [44]. During mammalian oogenesis, maternal mRNAs are synthesized and accumulated to support the meiotic maturation of oocytes [44]. mRNA is degraded at the stage of meiotic resumption, and about 90% of maternal mRNA is eliminated during MZT [45]. After the onset of mRNA elimination, the zygotic genome is activated from the stage of zygote formation to a 2-cell embryo [46]. The first and main pathway for mRNA degradation involves the shortening of the poly(A) tail, known as deadenylation [45]. An important enzyme complex involved in the shortening of the poly(A) tail is CCR4-NOT deadenylase. CCR4-NOT does not directly interact with mRNA. The CCR4-NOT adapter that attracts CCR4-NOT to target mRNAs is the BTG/TOB protein family, which includes the BTG4 protein [44]. As a result of impaired BTG4 function, there is a significant delay in the degradation of maternal mRNA during MZT. Pathogenic variants in the *BTG4* gene lead to a disruption of the interaction of BTG4 with the CCR4-NOT complex and the arrest of the first zygote cleavage as a result of the accumulation of maternal mRNAs [47]. Btg4-null female mice produce morphologically normal oocytes that arrest at the single-cell stage and are infertile, while males show normal fertility [43]. Zheng et al. identified homozygous pathogenic variants in the *BTG4* gene associated with the ZCF phenotype in four unrelated women [47].

## 4. Genes of SCMC Components

Several MEGs encode components of the subcortical maternal complex (SCMC), a multiprotein complex that is uniquely expressed in the oocytes and early embryos and plays a vital function(s) in early embryo development [48]. There are a number of genes encoding the components of the SCMC, including the *KHDC3L*, *NLRP2*, *NLRP5*, *NLRP7*, *PADI6*, *TLE6*, and *OOEP* genes [49]. They are involved in spindle assembly formation and positioning by controlling the formation of F-actin cytoskeleton and may be involved in the regulation of maternal to zygotic translation and epigenetic reprogramming. In humans, pathogenic variants in SCMC genes cause various conditions, including early embryonic developmental arrest [50].

The ***KHDC3L*** (*KH domain containing 3-like*) gene is located on 6q13. It consists of three exons and encodes for a 217-amino acid protein. *KHDC3L* is expressed in several tissues and is a member of SCMC [51]. Homozygous or compound-homozygous variants in the *KHDC3L* gene are the main cause of a biparental complete hydatidiform mole (BiCHM, a recurrent familial hydatidiform mole), a rare gestational abnormality, characterized by trophoblast overgrowth and the absence of embryo development. *KHDC3L* variants are found in 5–10% of BiCHM cases and are associated with the widespread loss of methylation (LoM), of germline differentially methylated regions (gDMRs), and of imprinted genes in hydatidiform mole material [52]. KHDC3L transcripts are expressed in a variety of human tissues, including all oocytes stages and preimplantation embryos. To date, different variants have been reported in *KHDC3L* in cases of early pregnancy loss and hydatidiform moles [51,53,54].

The ***NLRP2*** (*NLR family, pyrin domain containing 2)* gene is located on 19q13.42. It consists of 13 exons and encodes for a 1062-amino acid protein. *NLRP2* is expressed in several tissues, such as lung, placenta, thymus, and is a member of SCMC. Pathogenic variants in *NLRP2* have been described in one family with a recurrent Beckwith–Wiedemann syndrome due to the loss of DNA methylation at the imprinted KvDMR1 CpG island (CGI) [55]. Homozygous and compound-heterozygous variants in *NLRP2* were found in human embryonic arrest (2- to 7-cell stage) or fertilization failure. Phenotype variability can be explained by the fact that different variants impair the function of protein to different extents. Nlrp2-deficient male mice had normal fertility [50].

The ***NLRP5*** (*NLR family, pyrin domain containing 5*) gene is located on 19q13.43. It consists of 15 exons and encodes for a 1200-amino acid protein. *NLRP5* is expressed in several tissues with primary expression in germ cells and preimplantation embryos [56]. Homozygous and compound-heterozygous variants in the *NLRP5* gene associated with a phenotype of embryonic developmental arrest (2- to 7-cell stage), fertilization failure, or multi-locus imprinting disturbance (MLID) [57,58].

The ***NLRP7*** (*NLR family, pyrin domain containing 7*) gene is located on 19q13.42. It consists of 11 exons and encodes for a 1037-amino acid protein with predominant expression in germ cells and preimplantation embryos. To date, more than 60 pathogenic variants in *NLRP7* have been described [59]. Maternal homozygous and compound-heterozygous pathogenic variants in *NLRP7* gene are the most frequent causes of BiHM (75%), determining its role in the process of genomic imprinting [60].

The ***PADI6*** (*peptidyl arginine deiminase, type 6*) gene is located on 1p36.13. It consists of 17 exons and encodes for a 694-amino acid protein. *PADI6* is mainly expressed in the oocytes, mature sperm, and early embryos. *PADI6* is necessary for the activation of the embryonic genome. Padi6-deficient mice are sterile due to arrested embryonic development prior to the 4-cell stage due to failure to undergo zygotic genome activation (ZGA). Pathogenic loss-of-function variants in *PADI6* have been also described in the mothers of children with sporadic Beckwith–Wiedemann syndrome with MLID [61]. Biallelic variants in *PADI6* cause female infertility characterized as early embryonic developmental arrest, zygotic cleavage failure, and recurrent hydatidiform moles [62,63,64,65,66]. An in vitro expression study showed a significant decrease in PADI6 in mutation models, which may disturb the stability of the SCMC complex [62].

The ***TLE6*** (*transducin-like enhancer of split 6*) is located on 19p13.3. It consists of 17 exons and encodes for 572-amino acid protein. The absence of TLE6 results in asymmetric cleavage and early embryonic developmental arrest [67]. A functional study showed that the *TLE6* variants impaired the PKA-mediated phosphorylation of TLE6 and prevented the interaction between TLE6 and other SCMC proteins [68]. In mouse models, the knockout of the Tle6 gene has no effect on oogenesis but results in embryonic developmental arrest at the 2-cell stage, leading to female mouse infertility [48,67]. Homozygous and compound-heterozygous variants in the *TLE6* gene are associated with various phenotypes, including fertilization failure, early cleavage failure, embryonic developmental arrest on day 3, and a high rate of embryo fragmentation due to the disruption of F-actin and spindle assembly [14,66,68,69]. A woman with a heterozygous variant in *TLE6* had normal fertility and two healthy children [69]. In men with homozygous variants in *TLE6* fertility is preserved [14].

The ***OOEP*** (*oocyte-expressed protein*) is located on 6p13. It consists of three exons and encodes for a 149-amino acid protein. *OOEP* is mainly expressed in the oocytes and early embryos. The *OOEP* gene, similar to other genes of the complex SCMC, contributes to the symmetrical division of the zygote and is involved in the activation of the zygotic genome and also in the process of reparation DNA double-strand breaks (DSBs) [70]. Biallelic variants in *OOEP* lead to different phenotypes, including early embryonic developmental arrest, MLID [17,71].

## 5. Genes Involved in the Formation of the Zona Pellucida and Associated Disorders

The zona pellucida (ZP) is a glycoprotein matrix that is critical for oocyte maturation, ovulation, and fertilization through the induction of acrosome reaction, prevention of polyspermy, and preimplantation embryo development [72,73]. Human ZP is closed to rat ZP and also consists of four glycoproteins, called *ZP1*, *ZP2*, *ZP3*, and *ZP4* (Table 1), encoded by the corresponding genes [74]. In mammals, *ZP* genes are expressed exclusively in growing oocytes [75]. During follicular development, the *ZP* genes are activated, producing the ZP glycoproteins to form the zona pellucida [76]. ZP2-ZP3 dimers polymerize to form the long fibrils cross-linked by ZP1 that constitute the thick extracellular matrix.

Pathogenic variants in human *ZP1*, *ZP2,* and *ZP3* genes result in a thinning or absent ZP, empty follicle syndrome, or abnormal oocyte formation, which leads to female infertility [77]. The eggs of the *ZP1*^MT/MT^ female rat were not surrounded by a ZP and lost their fertilization capacity [78]. Mutant *ZP1* secreted neither protein ZP1 nor proteins ZP3 and ZP4 and the interaction with ZP2 was disrupted, which eventually led to the absence of zona pellucida formation, suggesting that normal ZP1 protein is crucial for the ZP formation and for natural fertilization [78]. Heterozygous or compound-heterozygous variants in *ZP1* described in patients with several IVF/ICSI failure and who lacked a ZP. Heterozygous *ZP4* gene variants are not associated with infertility due to the lack of ZP [79].

Functional studies reveal that biallelic pathogenic variants impair the assembly of the ZP proteins and therefore result in oocyte degeneration [77]. Many researchers have not found a relationship between the pathogenic variants in *ZP4* and infertility [51,53]. The ZP morphology in Zp4−/− rats was normal [80]. Thus, to date, there is not enough data on the direct role of the *ZP4* gene in the etiology of disturbance in the formation of the ZP.

## 6. Other Genes Associated with Female Infertility

Some and other genes have been found to lead to female infertility associated with oocyte maturation defects, early embryonic development arrest or recurrent hydatidiform mole.

The ***WEE2*** (*WEE1 homolog 2*) gene is located on 7q34. It consists of 12 exons and encodes for 567-amino acid protein. WEE2, or Wee1B, belongs to the WEE kinase oocyte-specific protein family [15]. WEE2 is a tyrosine kinase that acts as a key regulator of meiosis during prophase I (GV stage) and metaphase II [81]. At GV stage it involves in maintaining meiotic arrest by inhibiting the maturation-promoting factor (MPF) through inactivating CDK1 [82]. In the MII stage, *WEE2* is essential for MII exit. It was found that with reduced WEE2 expression in human and mouse oocytes, high MPF activity was noted, which led to impaired exit from MII and, as a result, the formation of pronuclei, which is an indicator of fertilization failure [15]. Many researchers show that homozygous or compound heterozygous variants in *WEE2* lead to female infertility due to fertilization failure [15,83,84,85,86]. The oocytes of these patients were morphologically normal; after ICSI, these oocytes were able to extrude the second polar body but did not form a zygote. Functional studies showed that pathogenic variants in *WEE2* significantly decrease the amount of WEE2, resulting in impaired WEE2 and CDK1 phosphorylation, leading to MII termination and subsequent fertilization failure due to OMA at MII [15,83,84,85,86]. Researchers have also shown that the injection of *WEE2* cRNA into female oocytes resulted in in vitro blastocyst formation on day 6 [15]. Preimplantation genetic testing showed that they had a normal karyotype, which gives the potential for the further use of this method in patients with pathogenic variants in the *WEE2* gene [15].

The ***PANX1*** (*pannexin 1*) gene located on 11q21 consists of five exons and encodes for a 422-amino acid protein, pannexin 1. PANX1 belongs to the pannexin family of glycoproteins, which consists of three members, PANX1, PANX2, and PANX3. They form single membrane channels that provide the molecular exchange between the cell cytoplasm and the extracellular environment, including the transport of adenosine 5′-triphosphate (ATP) [87,88]. In humans, *PANX1* is ubiquitously expressed and has recently been identified in the female reproductive system, including oocytes and embryos at the 2- to 8-cell stage [89]. Homozygous or compound-heterozygous pathogenic variants in *PANX1* lead to the release of more ATP into the extracellular space and to oocyte degeneration [90]. The most oocytes remain immature and degenerated before or right after fertilization [89]. Heterozygous *PANX1* variants were found to be responsible for the development of these phenotype, termed “oocyte death syndrome” [91]. Functional studies showed that pathogenic variants in *PANX1* gene disrupt the process of *PANX1* glycosylation and result in disturbed channel activity [91]. These data confirm that impaired glycosylation processes and channelopathy lead to female infertility. In men, heterozygous variants in *PANX1* gene do not lead to male infertility, proving their important function for normal oogenesis and oocyte maturation.

The ***REC114*** (*meiotic recombination protein 114*) gene located on 15q24 consists of six exons and encodes for 266-amino acid protein. The programmed formation of DSBs initiates the meiotic chromosome recombination that is essential for proper chromosome segregation during meiosis I. There are a number of genes that control the process of DSBs formation and are essential for oocyte meiosis and early embryonic development including *IHO1*, *MEI4*, and *REC114* [92]. MEI4 forms a stable complex with REC114 and IHO1. In *S. cerevisiae* model it was shown that these three proteins are expressed from the beginning of the prophase I of meiosis and the expression decreases after the synapsis of homologous chromosomes at the pachytene stage [93]. Homozygous or compound-heterozygous variants in the *REC114* gene are responsible for oocyte maturation arrest and early embryonic arrest [94]. Researchers also show that homozygous pathogenic *REC114* variants affecting the splicing process might cause a recurrent hydatidiform mole [95].

**Table 1 genes-13-01920-t001:** Maternal effect genes (MEGs) and other genes affecting female fertility in humans.

Gene Symbol	Locus	Function	Phenotype	OMIM	Inheritance	References
*TUBB8* *CDC20* *TRIP13*	10p15.3 1p34.2 5p15.33	Oocyte meiotic spindle assembly Components of spindle assembly checkpoints	OMA, IF, FF, EEDA, ZCF	616768 603618 604507	AD, AR AR AR	[21,24,32,35,37,38,96]
*4242PATL2*	15q21.1	mRNA binding/inhibition of post-transcription translation	OMA, IF, FF, EEDA	614661	AR	[40,97]
*BTG4*	11q23.1	Maternal mRNA decay	ZCF	605673	AR	[47]
*KHDC3L* *NLRP2* *NLRP5* *NLRP7* *PADI6* *TLE6* *OOEP*	6q13 19q13.42 19.13.43 19q13.42 1p36.13 19p13.3 6q13	SCMC member SCMC member SCMC member SCMC member SCMC member SCMC member SCMC member	EEDA, RHM, RPL EEDA, MLID EEDA, MLID EEDA, RHM EEDA, ZCF, RHM, MLID EEDA, IF EEDA, MLID	611687 609364 609658 609661 610363 612399 611689	AR AR AR AR AR AR AR	[50,51,58,68,71,98,99,100]
*ZP1* *ZP2* *ZP3* *ZP4*	11q12.2 16p12.3-p12.2 7q11.23 1q43	Formation of zona pellucida (ZP)	EFS, AZPF	195000 182888 182889 613514	AR AD, AR AD AR	[77,101,102,103,104,105]
*WEE2*	7q34	Regulator of meiosis during Prophase I and Metaphase II	OMA, FF	614084	AR	[106,107]
*PANX1*	11q21	Oocyte maturation	ODP	608420	AD	[90,108]
*REC114*	15q24.1	Initiation of double-strand breaks and homologous recombination of DNA	ODP/EEDA, RHM	618421	AR	[94,109]

AR—autosomal recessive; AZPF—abnormal zona pellucida formation; EEDA—early embryo developmental arrest; EFS—empty follicle syndrome; FF—fertilization failure, MLID—imprinting disease; IF—implantation failure; ODP—oocyte death phenotype; OMA—oocyte maturation arrest; RHM—recurrent hydatidiform moles; RPL—recurrent pregnancy loss; ZCF—zygotic cleavage failure.

## 7. mtDNA, Oocyte Maturation and Embryo Developmental Competence

In humans, male and female gametes are not completely equal in their contribution genetic material to embryo. Beside the maternal haploid nuclear genome, oocytes also Deliver mitochondrial genome to the embryo; therefore, various defects in the mitochondrial genome are maternally inherited except in unique cases of genetic disorders resulting from mitochondrial DNA (mtDNA) mutations transmitted by sperm [110,111] or the biparental inheritance of mtDNA [112].

Mitochondria provide the cells with the adenosine triphosphate (ATP) production via oxidative phosphorylation (OXPHOS). In mammalian, circular full-length mtDNA molecule contains 16,569 base pairs and consists 37 genes, including 13 protein coding (eleven subunits of the electron transport chain and two subunits of ATP synthase) genes, 22 genes coding tRNAs, and two rRNA genes [113]. Mitochondria and mtDNAs are the factors involved in human aging. The mitochondrial genome lacks nucleosomes [114], and is therefore highly susceptible to genotoxic damage, especially to oxygen-free radicals, reactive oxygen species (ROS). The mutation rate of the mitochondrial DNAs is about 20 times higher than the nuclear genome. Higher mutation rates in mitochondrial genome are due to less effective mtDNA repair mechanisms and the lack of histones, which protected mtDNAs; therefore, the mitochondrial genome does not have such opportunities for DNA reparation [115]. In addition, mutations of the mitochondrial genome are facilitated by the fact that more oxygen-free radicals are formed in the mitochondria as a result of OXPHOS.

The role of mitochondria and mitochondrial genome in mammalian fertility, especially, gamete and embryo quality, embryo developmental competence has been shown in many studies [116,117,118,119]. These studies have found a correlation between the amount of mtDNA and the amount of adenosine triphosphate in the oocytes and the morphology and fertility of female gametes, as well as between ATP content in oocytes and morphology and the development potential of preimplantation embryos. This is due to the fact that the energy potential in the human embryo is determined exclusively by the functional competence of oocyte mitochondria, which produce ATP.

Early human embryos are characterized by the absence of a replication and transcription of the mitochondrial genome. The resumption of mtDNA transcription in the human embryo begins simultaneously with the activation of its own genome, which corresponds to 72 h after fertilization. mtDNA replication begins during preimplantation development at the blastocyst stage, when embryos require adequate energy levels for successful division [120]. Existing evidence suggests that the correct function of oocyte mitochondria is of decisive importance in the early stages of embryogenesis. Therefore, defects in mitochondria and mitochondrial genome as a cause of reproductive aging and decreasing female fertility is very attractive [121].

During mammalian oogenesis, the number of mtDNA molecules significantly increases in the germ cells, especially at late stages of the meiosis. So, primordial germ cells (PGCs) contain approximately 200 mtDNA copies, and mature oocytes possess as many as 150,000–400,000 mtDNAs, about an order of magnitude greater than their number in most human somatic cells. The dramatic increasing of mtDNA content in the female germ line during post-pubertal folliculogenesis and the further decreasing in early embryogenesis suggest that it may account for the bottleneck effect of mtDNA inheritance in humans [118]. The mtDNA copy number in oocytes and polar bodies shows both decreased and increased mtDNA in oocytes. Female gametes with low mtDNA content demonstrate poor embryo developmental competence; therefore, the measurement of mtDNA copies in the oocyte’s genome provides additional information about oocyte quality and embryo developmental competence [121]. It was stated that for normal embryonic development, the critical threshold is 40,000–50,000 mtDNA copies in the oocyte [119]. A large number of mitochondrial DNA molecules in a mature oocyte makes it possible to provide mitochondria and mtDNAs to the cells of preimplantation human embryos before the resumption of mtDNA replication and mitochondrial biogenesis. It should be noted that the metabolism in human oocytes and embryos differs from the metabolism in somatic cells. Early embryos do not receive ATP from OXPHOS but rather as a result of aerobic glycolysis, the so-called Warburg effect [122]. Mitochondria in oocytes and preimplantation embryos have a small number of crysts, and the oxygen consumption remains exceptionally low until the blastocyst stage [123].

Oogenesis in older women is characterized by prolonged arrest in the prophase I, which results in age-related changes accumulated in human genome. The inherent mitochondrial genome instability and the prolonged exposure of mtDNA molecules to a reactive oxygen species and make it susceptible to various genome mutations; therefore, many germ cells harbor mtDNA mutations, especially common rearrangements, for example, a 4.9 kb deletion [116,119]. These in turn increase mtDNA copy number as a compensatory response. Although, generally, the proportion of mutant mtDNA is quite low relative to the total mtDNA content in mature oocytes (a low level of heteroplasmy) [119]. It has been hypothesized that even low levels of mtDNA mutations, and also aneuploidy, in turn increase the mtDNA copy number in oocytes as a compensatory response. Deficiencies in ATP production in oocyte, apparently, can impair meiotic spindles and negative affect chromosome segregation, with increasing aneuploidy commonly resulting in oocyte or embryo death.

It is thought that the purifying selection of most deleterious mtDNA mutations in oocytes is responsible for their elimination during mammalian oogenesis. The studies on mutant mice showed that many mtDNA variants undergo negative selection in germ-line cells and are not inherited [115,124]. So, although somatic mutations in mtDNAs in PolG^mut/mut^ mice show high frequency, and the frequency of the deleterious variants of mitochondrial genome in their offspring is relatively low [115,118]. It is thought that the negative selection of germ cells with abnormal mtDNA is responsible for the elimination of most of the deleterious mutations of mitochondrial genome in mammalian embryogenesis. It implies that there is some purifying selection which eliminates oocytes with mtDNA mutations. Indeed, the vast majority of deleterious variants of the mitochondrial genome are negatively selected in oogenesis, showing a germ-line bottleneck in their transmission to offspring [115]. A number of genetic studies showed that some variants of mitochondrial genes can cause pregnancy loss. So, deleterious mutations in OXPHOS protein-coding genes are associated with reproductive failure in mice [117,125]. A prevalence of mtDNA variants in MT-ND1 and the D-loop region detected in the oocytes of patients with recurrent pregnancy loss (RPL) suggest that mtDNA mutations are one of genetic factors damaging embryonal development [117,119,125]. However, in contrast to strong purifying selection against deleterious protein-coding mtDNA mutations, negative selection against some tRNA and rRNA gene variants is less distinct, apparently due to their mild pathogenicity and/or low heteroplasmy [119]. Thus, the content of mtDNA copies and some their gene variants in oocytes and/or follicular cells can be related to the quality of the oocytes and the frequency of aneuploidy that dramatically affect the ability of the embryo to further develop and, thus, the progression of pregnancy. Females with a small number and/or quality of mtDNA molecules in their gametes have lower ovarian reserves and decreased fertility. This is a predictor for poor developmental potential and implantation of an embryo, leading to a higher risk of early pregnancy loss.

## 8. Genes Involved in Oocyte Maturation Defects and Early Embryo Lethality as Cause of IVF/ICSI Failures

A significant proportion (more than 50–60%) of IVF/ICSI cycles are ineffective for achieving and developing pregnancy using assisted reproduction techniques (ART). Many infertile couples have experienced the recurrent failure of ART programs caused by low quantity and/or quality of gametes, morphology, and genetic defects detected in preimplantation embryos. Therefore, preserved gametogenesis, the quality of gametes and the early stages of reproduction, and embryonic development are essential components of human fertility. In reality, the sufficient quantity and quality of gametes is the most important factor in the determination of both successful in vivo and in vitro fertilization and pregnancy progression. To date, there are no highly predictive indicators that allow the assessment of the quality of oocytes beside its morphology. Recently, a number of studies have revealed some molecular and genetic factors as potential markers for assessing the quality of female gametes and allowing the election of embryos with development competence (Table 1).

On the one hand, genetic infertility factors are common in patients with severe gametogenesis defects and/or gamete abnormalities, idiopathic infertility, and unsuccessful repeated IVF/ICSI procedures. So, single women with oocyte maturation defects, in particular because of pathogenic *NLPR2* variants, gave birth after several unsuccessful IVF/ICSI programs, were reported in the literature [56]. A similar situation was mentioned in patients with a recurrent hydatidiform mole. It was found that only 1% of female patients carrying the biallelic pathogenic variants of *NLRP7* gene were able to give birth from spontaneous conceptions [59], and most of these patients needed an IVF/ICSI procedure with donor oocytes. Thus, the search and detection of pathogenic variants in fertility-related genes may be essential for genetic counseling and the choice of tactics for solving reproduction problems.

On the other hand, many successful and unsuccessful IVF/ICSI procedures provide unique opportunities for the analysis of infertility causes and pathogenic mechanisms related to fertilization failures, defects of oocyte maturation, and early embryonic development. Whole-exome sequencing (WES) and whole-genome sequencing (WGS), transcriptome and proteome analysis, and the complex evaluation of morphology and various genetic and epigenetic factors responsible for gamete quality and embryonal development competence allows the improvement the diagnosis, genetic consulting, treatment, and successful solving of reproductive problems, especially regarding the efficacy of ART programs. These studies allow the revelation of novel fertility-related genes and uncover their functions and the signaling pathways, functions, and molecular mechanisms involved in human reproduction.

## 9. Conclusions

Oocyte maturation is a well-organized complex process that includes meiotic division and recombination, nuclear maturation, and epigenetic modification. Each stage of this process is regulated by a large network of genes. Pathogenic variants in these genes can result in the recurrent failures of IVF/ICSI programs due to a poor response to ovarian stimulation, oocyte maturation arrest, poor quality of gametes, fertilization failure, or early embryonic development arrest. Knowledge regarding these genes and their variants, coding products, and functions, which are critical for human reproduction, is essential both for the diagnosis of genetically determined forms of female infertility and for the successful solution of the problem of childbearing, especially when utilizing assisted reproduction techniques, as well as for the prevention of genetic disorders in offspring.

## Figures and Tables

**Figure 1 genes-13-01920-f001:**
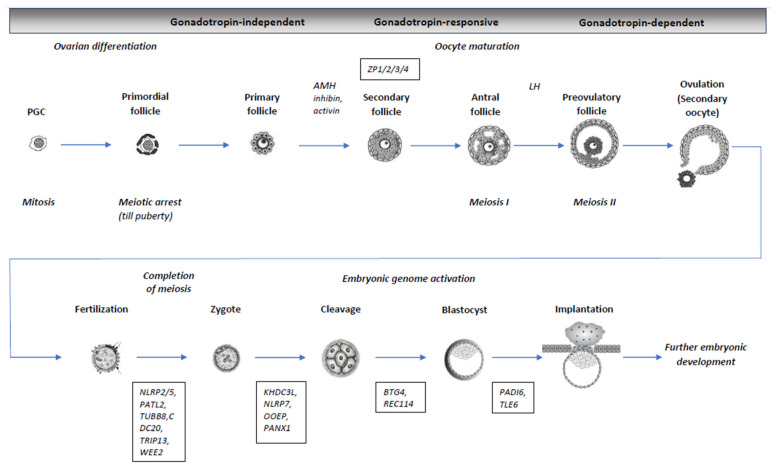
Schematic representation of oogenesis and folliculogenesis in humans, ovulation, fertilization, and early stages of embryonic development with indication of stages, hormones, and some genes involved in their regulation. Gonadotropin-independent, gonadotropin-responsive, and gonadotropin-dependent stages are indicated above. Oocyte maturation period is outlined with a dotted line. PGC, primordial germ cell (gonocyte). The genes are listed in rectangular boxes. AMH—Anti-Müllerian hormone; LH—Luteinizing hormone.

## Data Availability

Not applicable.

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
