# Peer review of "Genetics of Oocyte Maturation Defects and Early Embryo Development Arrest"

_genes, 2022, doi:10.3390/genes13111920_

Round 1

Reviewer 1 Report

In this manuscript, Solovova and Chernykh reviewed the existing literature about the pathogenic factors that interfere with oocyte maturation and cause wide range of infertility. This study covers a good number of genes. However, the manuscript is very terse and, in several points, it lacks appropriate references. 

Here are some minor comments, that authors need to go through them:

The aim/objective of the study is not mentioned in the abstract.

Page 1, lines 31-38 and 40-45: Authors need to cite appropriate references. No references are present for these two paragraphs.

Page 2, lines 48-53: Reference number 4 is not appropriate at all. 

Page 2, lines 57-59 and 66-80 and 81-84: Authors need to cite appropriate references. Only one appropriate work is cited within the whole page.

Figure 1: This figure is too crowded. There are many genes mentioned in the boxes but there is no explanation or reference for these genes in the text. Conversely, some genes are discussed in the text but not mentioned in the figure. Also, the antral follicle is in first meiosis not second meiosis. Authors need to talk about the genes within the text or remove them. 

Page 3, lines 94-99: No references either in text or table 1. 

Table 1: Authors need to verify whether all the listed genes are known as maternal effect genes in human. A maternal effect gene should be expressed in oocyte and persist during early embryonic development. Also, the appropriate reference should be cited for each gene. 

Page 6, lines 201-205: No references are cited. Also, mutations in genes encoding for SCMC members don’t cause fertilization failure.

Page 6, lines 230-235: Nlrp5 mutation also caused a liveborn with multi locus imprinting disorder (MLID). This study should be cited. 

Page 7: Ooep is another member of SCMC and mutations in it is associated with MLID. Nlrp9 is another member and mutations are associated with MLID and BiCHM. These studies should be cited and discussed.

Page 11, lines 462-465: No references are cited.

Author Response

Point 1: The aim/objective of the study is not mentioned in the abstract.

Response 1: Thank you very much for your comment! we have corrected the abstract taking into account your recommendations.

Point 2: Page 1, lines 31-38 and 40-45: Authors need to cite appropriate references. No references are present for these two paragraphs.

Response 2: Thank you for the comment. We cited this lines using the appropriate reference.

Point 3: Page 2, lines 48-53: Reference number 4 is not appropriate at all.

Response 3:  Thank you! We have corrected this part.

Point 4: Page 2, lines 57-59 and 66-80 and 81-84: Authors need to cite appropriate references. Only one appropriate work is cited within the whole page.

Response 4: Thank you for this comment. We have made the appropriate references.

Point 5: Figure 1: This figure is too crowded. There are many genes mentioned in the boxes but there is no explanation or reference for these genes in the text. Conversely, some genes are discussed in the text but not mentioned in the figure. Also, the antral follicle is in first meiosis not second meiosis. Authors need to talk about the genes within the text or remove them. 

Response 5: Due to the insertion of the boxes with genes, positions on the picture were changed. Thank you very much for noticing this! We have fixed all comments, removed genes that were not described in the text and added genes reviewed in this paper.

Point 6: Page 3, lines 94-99: No references either in text or table 1. 

Response 6: Thank you very much for the comment! We have corrected this part.

Point 7: Table 1: Authors need to verify whether all the listed genes are known as maternal effect genes in human. A maternal effect gene should be expressed in oocyte and persist during early embryonic development. Also, the appropriate reference should be cited for each gene. 

Response 7: Really, not all of these genes are MEGs, thank you for important comment. We decided not to separate information about all genes and leave it in one table, the table was renamed. We also cited reference for each gene.

Point 8: Page 6, lines 201-205: No references are cited. Also, mutations in genes encoding for SCMC members don’t cause fertilization failure.

Response 8: Thank you for this comment. We have made the appropriate corrections.

Point 9: Page 6, lines 230-235: Nlrp5 mutation also caused a liveborn with multi locus imprinting disorder (MLID). This study should be cited. 

Response 9: Thank you for this comment. We have made the appropriate corrections.

Point 10: Page 7: Ooep is another member of SCMC and mutations in it is associated with MLID. Nlrp9 is another member and mutations are associated with MLID and BiCHM. These studies should be cited and discussed.

Response 10: Thank you very much for your comment! When writing the review, we did not pay attention to the importance and contribution of the OOEP gene. We have taken into account your remarks and presented a brief description of it among other genes of the complex SCMC. Unfortunately, we could not find information about the relationship of the NLRP9 gene with MLID and BiCHM, so we did not include it in the review article.

Point 11: Page 11, lines 462-465: No references are cited.

Response 11: Thank you! We cited this lines using the appropriate reference.

We thank Reviewer 1 for the extensive and careful work that was done for improving the quality of our manuscript.

Reviewer 2 Report

The abstract clearly illustrates the aims of this article. As for the title, it's informative enough to draw its potential reader's attention. It is known that genetic, epigenetic, endocrine, immunological etc. factors affect the functioning of the reproductive system and on human fertility.
The introduction of this article successfully explains why the current study is important and the clinical implications from pathogenic variants of the genes that control these processes are.
This article reviews the genes that play a very important role in folliculogenesis and oocyte maturation. Pathogenic variants in these genes as well as epigenetic factors can lead to poor oocyte quality, fertilization failure or stopping at early embryo development
1.    How are genes selected - a systematic search?
2.    Names of the genes need to be written in italics and with capital letters (line 55,56,57, 58…..171….258…). Correct in the text if there are such technical errors somewhere.
3.    The fourth, fifth and sixth chapters on genes are not accompanied by tables. If the authors follow the concept given in Figure 1, they could also form tables in these chapters (optional).
4.    MIM or OMIM numbers (https://www.omim.org/ )?  - line 58,59, 95, 98,99 and in Table 1
5.    Line 102  - not TBB8, but TUBB8
6.    Line 172 - ……” During mitosis, TRIP13 protein supports the spindle checkpoint silencing, and in meiosis TRIP13 is involved in recombination pathways”…...- lack of references?
As for the all references, they are relevant, related to the topic, the correctly cited. There are some articles that may be useful to the authors to complete and improve the data in the article
1.    In the sixth chapter, when the authors consider the PANX 1 gene, it can be mentioned that homozygous variants in PANX1 cause human oocyte death and female infertility (Wang, W., Qu, R., Dou, Q. et al. Homozygous variants in PANX1 cause human oocyte death and female infertility. Eur J Hum Genet 29, 1396–1404 (2021). https://doi.org/10.1038/s41431-020-00807-4).
2.    Lines 371-372 … .Various mitochondrial genome defects are maternally inherited, except for unique cases of genetic disorders resulting from mtDNA mutations transmitted by sperm… - references could be given
3.    References could be given in all tables.

The article is written very detailed and the figure/tables correctly illustrate the information. The reader understands the aims and the importance of the topic although the large volume of scientific data. Providing readers with possible hypotheses and solutions in the context of scientific facts would also be helpful.
All these refinements may make the article more interesting for readers from multiple backgrounds.

Author Response

Point 1: How are genes selected - a systematic search?

Response 1: A systematic search was conducted on the most studied human genes associated with impaired female fertility.

Point 2: Names of the genes need to be written in italics and with capital letters (line 55,56,57, 58…..171….258…). Correct in the text if there are such technical errors somewhere.

Response 2: Thank you for a comment. Human genes were written in italics and with capital letters. In these lines (line 55,56,57, 58…..171….258…) we described mouse genes, so they were written in this format.

Point 3: The fourth, fifth and sixth chapters on genes are not accompanied by tables. If the authors follow the concept given in Figure 1, they could also form tables in these chapters (optional).

Response 3:  Thank you very much for the comment. We decided not to separate information about all genes and leave it in one table, we inserted a table after describing all the genes.

Point 4: MIM or OMIM numbers (https://www.omim.org/ )?  - line 58,59, 95, 98,99 and in Table 1.

Response 4: Thank you for the comment. We have made the appropriate corrections.

Point 5: Line 102  - not TBB8, but TUBB8.
Response 5: Thank you. We fixed a mistake.

Point 6: Line 172 - ……” During mitosis, TRIP13 protein supports the spindle checkpoint silencing, and in meiosis TRIP13 is involved in recombination pathways”…...- lack of references?

Response 6: Thank you very much for the comment. We have corrected this part.

Point 7: In the sixth chapter, when the authors consider the PANX 1 gene, it can be mentioned that homozygous variants in PANX1 cause human oocyte death and female infertility (Wang, W., Qu, R., Dou, Q. et al. Homozygous variants in PANX1 cause human oocyte death and female infertility. Eur J Hum Genet 29, 1396–1404 (2021). https://doi.org/10.1038/s41431-020-00807-4).
Response 7: Thank you, we have made the appropriate corrections.

Point 8: Lines 371-372 … .Various mitochondrial genome defects are maternally inherited, except for unique cases of genetic disorders resulting from mtDNA mutations transmitted by sperm… - references could be given.

Response 8: Thank you for this comment. We have made the appropriate corrections.

Point 9: References could be given in all tables.

Response 9: Thank you. We cited Table 1 using the appropriate references.

Point 10: Providing readers with possible hypotheses and solutions in the context of scientific facts would also be helpful.

Response 10: Thank you very much for these noticing. In Conclusion section we emphasized that the analysis of world literature data will be useful in improving the diagnosis in patients with impaired oocyte maturation and female factor infertility, leading to failure of IVF/ICSI programs due to poor quality of oocytes.

We thank Reviewer 2 for the extensive and careful work that was done for improving the quality of our manuscript.